# COVID-19 Mimics Pulmonary Dysfunction in Muscular Dystrophy as a Post-Acute Syndrome in Patients

**DOI:** 10.3390/ijms24010287

**Published:** 2022-12-24

**Authors:** Suresh C. Tyagi, Sathnur Pushpakumar, Utpal Sen, Sri Prakash L. Mokshagundam, Dinesh K. Kalra, Mohamed A. Saad, Mahavir Singh

**Affiliations:** 1Department of Physiology, School of Medicine, University of Louisville, Louisville, KY 40202, USA; 2Division of Endocrinology, Metabolism and Diabetes and Robley Rex VA Medical Center, University of Louisville School of Medicine, Louisville, KY 40202, USA; 3Division of Cardiovascular Medicine, University of Louisville School of Medicine, Louisville, KY 40202, USA; 4Division of Pulmonary, Critical Care and Sleep Disorders Medicine, University of Louisville School of Medicine, Louisville, KY 40202, USA

**Keywords:** lung dysfunction, coronavirus infection, multiorgan damage, disease management, neopterin

## Abstract

Although progressive wasting and weakness of respiratory muscles are the prominent hallmarks of Duchenne muscular dystrophy (DMD) and long-COVID (also referred as the post-acute sequelae of COVID-19 syndrome); however, the underlying mechanism(s) leading to respiratory failure in both conditions remain unclear. We put together the latest relevant literature to further understand the plausible mechanism(s) behind diaphragm malfunctioning in COVID-19 and DMD conditions. Previously, we have shown the role of matrix metalloproteinase-9 (MMP9) in skeletal muscle fibrosis via a substantial increase in the levels of tumor necrosis factor-α (TNF-α) employing a DMD mouse model that was crossed-bred with MMP9-knockout (MMP9-KO or MMP9^-/-^) strain. Interestingly, recent observations from clinical studies show a robust increase in neopterin (NPT) levels during COVID-19 which is often observed in patients having DMD. What seems to be common in both (DMD and COVID-19) is the involvement of neopterin (NPT). We know that NPT is generated by activated white blood cells (WBCs) especially the M1 macrophages in response to inducible nitric oxide synthase (iNOS), tetrahydrobiopterin (BH4), and tetrahydrofolate (FH4) pathways, i.e., folate one-carbon metabolism (FOCM) in conjunction with epigenetics underpinning as an immune surveillance protection. Studies from our laboratory, and others researching DMD and the genetically engineered humanized (hACE2) mice that were administered with the spike protein (SP) of SARS-CoV-2 revealed an increase in the levels of NPT, TNF-α, HDAC, IL-1β, CD147, and MMP9 in the lung tissue of the animals that were subsequently accompanied by fibrosis of the diaphragm depicting a decreased oscillation phenotype. Therefore, it is of interest to understand how regulatory processes such as epigenetics involvement affect DNMT, HDAC, MTHFS, and iNOS that help generate NPT in the long-COVID patients.

## 1. Introduction

The global experience and knowledge about COVID-19 (a clinical syndrome post SARS-CoV-2 infection) keep improving towards identification of certain patterns of SARS-CoV-2 mediated pathogenic features in various organs/tissues of the patients such as blood vessels (vasculature), muscles, retinae, hearts, kidneys, lungs, and the joints, however; the researchers are continuously striving to help design appropriate interventional strategies to mitigate the deleterious effects in patients who are commonly referred as the ‘COVID-19 long-haulers. Many deleterious effects in survivors are likely caused by the hyperinflammatory host response (i.e., cytokine storm), that may lead to prothrombotic state rather than direct toxicity by SARS-CoV-2 encoded gene product(s). Unfortunately, in the worst-case scenario, the obvious outcome leads to multisystem or multi-organ dysfunction including respiratory failure, and hypercoagulability phenotype which usually results in critical illness, and death [1,2,3,4]. Further, observations from the world over have suggested that survivors continue to have/exhibit symptoms that may persist or occur months after the initial SARS-CoV-2 infection (generally referred to as a post-COVID condition or long-COVID). Authors strongly believe that findings originating from COVID-19 related pathologic investigations, across a variety of modalities, can be extremely helpful not only for diagnosis, and treatment guidance but also for developing preventive tools/methodologies to tackle future disease conditions and similar viral outbreaks that disproportionately affect people who have pre-existing conditions such as Duchenne muscular dystrophy (DMD). These patients often face high risk of severe disease from SARS-CoV-2 infection since COVID-19 induced restrictive lung function (because of low lung volume), use of corticosteroids in DMD patients (that usually results in immunosuppression), and co-morbidity such as obesity, and hypertension may essentially contribute to severe disease outcomes [5,6,7,8].

DMD is an inherited condition and is referred as an X-linked genetic disease. Briefly, it is a severe, progressive, and muscle-wasting condition that leads to difficulties with movement, and to assisted ventilation. The disease is most common in male children that are usually characterized by proximal muscle weakness followed by calf muscle hypertrophy. The patients become wheelchair-bound around 12 years of their age. Sadly, most of them die of cardiorespiratory complications in their late teens to early twenties or thirties. It is caused by a mutation(s) in the *DMD* gene that abolishes the production of the ‘dystrophin’ protein. The *DMD* gene is the largest known human gene that provides instructions for making the dystrophin protein which is located primarily in muscles that are primarily used for movement such as the skeletal muscle. Further, the cardiac muscle is also affected in DMD patients. It has been demonstrated that a small amount of dystrophin is also present inside the nerve cells of brain. In skeletal, and cardiac muscles, dystrophin serves as a part of the group of proteins (i.e., a protein complex) that work together to strengthen the muscle fibers thus protecting cells from injury since muscle cells contract and relax (meaning that dystrophin acts as an anchor, thus connecting each muscle cell’s structural framework; cytoskeleton with the lattice of proteins, and other molecules outside the cell, i.e., the extracellular matrix; ECM). It is also believed that the dystrophin complex also plays an important role(s) in cell signaling pathways by interacting with other proteins that send, and receive chemical signals; however, muscles without dystrophin are more sensitive to damage, thus resulting in progressive loss of the muscle tissue, its function, in addition to cardiomyopathy.

Recent studies have truly enhanced our understanding of the primary, and secondary pathogenetic mechanism(s) that take place in DMD-affected patients. It is worth mentioning here that the guidelines for the multidisciplinary care for DMD patients that address obtaining a genetic diagnosis and managing various aspects of the disease have been well-established in advanced countries. In addition, several therapies that aim to restore the missing dystrophin protein or address secondary pathology have already received regulatory approval. Nonetheless, a few novel therapeutic approaches are currently undergoing clinical development, since safer therapies are still needed to provide a cure for this devastating muscle disorder [9,10]. As mentioned earlier, DMD is one of the primary causes of skeletal muscle atrophy, and myopathy [11,12,13,14,15]. We know that the diaphragm muscle continuously supports cardiopulmonary function in a healthy individual by regulating inhalation, and exhalation processes; however, in DMD patients, the lack of dystrophin production leads to progressive diaphragm muscle (thoracic diaphragm) degeneration and necrosis along with respiratory muscles dysfunction (Figure 1).

## 2. Disarray in Cytoskeleton and Extra-Cellular Matric (ECM)

Mutation in the dystrophin protein, as previously mentioned, causes severe derangements in cytoskeletal, and ECM leading to activation of the innate immune system and subsequently prompting inflammation in the lung parenchyma. Since dystrophin is linked to intracellular cytoskeleton, as well as, to ECM maintenance; however, this link is lost during DMD disease progression causing diaphragm malfunctioning, and later the respiratory failure in DMD patients [16,17,18,19]. We believe that the diaphragm, being an isolated muscle structure, can be stimulated to synchronize better with other respiratory muscles to achieve better contraction, and thus to prevent or avoid its malfunctioning. ECM remodeling by its very nature implies the synthesis, and degradation of matrix proteins. Previous studies from our laboratory, using MMP9^-/-^ mice crossed with the DMD mice, revealed an important role of MMP9 in DMD etiology. In the lung, ECM composition is unique due to its higher elastin contents compared to that of collagen contents. However, during ECM remodeling, replacement of degraded elastin is slower as compared to rapid turnover of the collagen which, in turn, leads to fibrosis, stiffness, and obstructive respiration [20]. Previously, we also demonstrated that tumor necrosis factor-α (TNF-α) related weak inducer of apoptosis (TWEAK) induced MMP9 leading to skeletal muscle myopathy. Interestingly, neopterin (NPT), an inflammatory marker, is increased in several inflammatory disorders including in DMD patients [21,22,23,24] (Figure 1). NPT is generated by activated M1 macrophages via inducible nitric oxide synthase (iNOS), depleting tetrahydrobiopterin (BH4) and tetrahydrofolate (FH4) one-carbon metabolism (FOCM) via the epigenetic cycle. Further, FOCM is regulated by DNA methyl transferase (DNMT), and histone deacetylase (HDAC) [25,26]. We would like to point out that depletion of BH4 and FH4 causes oxidative stress through reactive oxygen species (ROS) generation, inflammasome formation, and MMP9 activation, all leading to changes in the diaphragm of DMD patients’ muscle towards atrophy [27]. The prominent changes in the muscle afflicted with dystrophin mutation are summarized in Figure 2. Concomitantly, in DMD patients, there is a loss of pulmonary endothelial cell barrier, and alveolar-epithelial, and interstitial barrier as the disease progresses over time. Together, these alterations can lead to blood-lung-barrier (BLB) leakage, thus resulting in pulmonary edema, and tissue congestion. Therefore, it is important to determine the mechanism(s) of BLB leakage, and pulmonary edema in DMD patients. ECM in the basement membrane consists of latent MMP/TIMP/NO (the “ternary complex”). During respiratory failure, the oxidative stress activates MMP, and that causes the inactivation of TIMP via peroxynitrite, and tyrosine/arginine nitrosylation [28]. Previously, our group has demonstrated the role of oxidative stress, and ROS involvement in the activation of Toll-like receptor 2/4 (TLR2/4) [29,30,31,32]. Others have shown that TLR4 can trigger an innate immune response that appears to be mediated by activated T-cells, infection, and pyroptosis (i.e., death by infection) [33,34]. Recent studies have demonstrated NLRP3 inflammasome formation in the skeletal muscle of DMD patients [35,36,37]. It is important to highlight that gasdermin D (GSDMD) is a protein that is specifically recruited by NLRP3 and is eventually cleaved by caspase-1 leading to pyroptosis [38,39]. Hence, it is important to investigate the mechanism(s) of inflammation by iNOS, and TLR4 activation during DMD disease progression [40,41,42]. In this context, we opine that treatment with HDAC3 (Sirt3) inhibitor might help mitigate early alveolar epithelial, and pulmonary capillary endothelial barriers disruption, and that may help preserve the blood-lung-barrier (BLB) functioning, and subsequently, that might help prevent leakage, and pulmonary edema/congestion in DMD patients [43,44,45,46]. Further, NOS, ROS, and cytokines deplete BH4 to produce NPT, that in turn leads to the activation of TLR and downstream signaling in the muscle tissue. Could inhibition of iNOS, and TLR4 suppress inflammatory cytokine storm (as seen in the COVID-19 patients), and thus mitigate DMD-induced diaphragm myopathy, remains to be seen?

## 3. Role of Inflammation and Epigenetics in the Generation of Neopterin (NPT)

Although muscular dystrophy disease is associated with a mutation in the dystrophin gene, it is not very clear whether this gene is stringently regulated by some kind of epigenetic programming or modifications for the *DMD* gene itself, and its regulatory sequence (promoter) or for that matter the participation of acetylation of the histone(s) proteins that are associated with the *DMD* gene. In this context, HDAC3 inhibitor (Sirt3) may be investigated to reveal the role (if any) played by the epigenetics machinery during muscle myopathy in the DMD patients. Since some studies have already shown the putative role(s) of DNA methyltransferases (DNMTs), and HDAC, however; the precise mechanism(s) has remained at large, so far. Furthermore, dysfunctional FOCM as incited by DNMT (*vide supra*), and the methylation modification via epigenetics regulation may also participate in the causation of muscle atrophy but again the details for the causation, and progression of DMD disease are also lacking [47,48,49,50,51]. In the past, we have shown an increase in TNF-α levels in skeletal muscle; however, it is unclear whether epigenetics controls the expression of immune response genes directly such as the molecules and cell types, namely the TNF-α, IL1, iNOS, Nox4, NPT, M1 macrophages, MMP9, and CD4^+^ and CD8^+^ cells during DMD disease pathogenesis [16]. We opine that Sirt3 inhibitor by virtue of its beneficial effects might mitigate some, if not all, abnormal expression levels of above the molecules that are supposedly responsible for diaphragm atrophy, and alveolar leakage from the lungs in the DMD patients, and during SARS-CoV-2 infection, and subsequent COVID-19 syndrome. Because inflammation, lung epithelial barrier dysfunction, multi-organ damage, congestive (cardio-pulmonary) failure, and the blood-lung barrier (BLB) leakage are important signature readouts during respiratory distress, thus understanding the mechanism(s) of BLB leakage is important to develop appropriate interventional strategies for the DMD patients [52,53]. A few studies have shown a robust increase in neopterin (NPT) in the DMD patients [22]. Interestingly, as mentioned earlier, NPT is generated by activated M1 inflammatory macrophages (M1) by inducible nitric oxide synthase (iNOS), tetrahydrobiopterin (BH4), and tetrahydrofolate (FH4) pathways. Some relevant investigations towards determining the levels of iNOS, Nox4 generated NPT, and oxidative peroxynitrite/nitrosylation levels, and activation of macrophage MMP-9 would be helpful too. Excessive NLRP3 inflammasome formation, and its sustained activation contribute to skeletal muscle inflammation, and associated injury in DMD patients [54,55]. As we know that gasdermin D (GSDMD) is a protein that is recruited by formation of the NLRP3 inflammasome for cleavage, and subsequent pyroptosis, and it is robustly elevated during DMD disease progression [56,57]. A recent study has demonstrated that inhibition of GSDMD reduced LPS-induced acute lung injury by reducing inflammation, and pyroptosis [58]. It is, therefore, important to determine the levels of M1 macrophages, inflammasome, TLR4/NLRP3/CD147, T helper, and T killer cells, and the mechanism(s) of innate immune activation that tend to cause endothelial BLB leakage during DMD disease progression. Furthermore, it would be a novel approach to determine whether the DMD pathogenetic changes induce DNMT2, and HDAC3 (Sirt3) levels. If that is the case then it may have an association with cytokine production involving the TNF-α, IL1β, iNOS, Nox4, NPT, M1 macrophages, and MMP-9 activation, and concomitant increase in the respective CD4^+^ and CD8^+^ cell populations. If that proves correct then Sirt3 inhibition (with inhibitor such as YC8-02, 3-TYP, etc.) might be able to mitigate their levels along with the reduction in diaphragm atrophy, and the lung tissue alveolar leakage [59,60,61,62,63,64].

## 4. Application of the Sirt Inhibitors in DMD Disease

Our unpublished study shows an increase in NPT, iNOS, Nox4, and TNF-α in the DMD mouse model. Earlier, we demonstrated an induction of MMP9 in DMD disease, thus, activation of MMP9 in DMD patients’ diaphragm, and lung is quite possible [16]. For example, when Sirt3 inhibition was tested, it seemed that it ameliorated the skeletal muscle microcirculation dysfunction in a hypoxic model [45]. Therefore, treatment with Sirt3 inhibitor might help mitigate the MMP9 activity, and levels of EMMPRIN (CD147). We, therefore; hypothesize that under such conditions the levels of TIMPs will be opposite to that of the MMPs, as an ‘in-built’ regulatory mechanism. In future, carefully designed pulmonary endothelial permeability experiments might be able to reveal alveolar leakage phenotype, and its subsequent mitigation by Sirt3 inhibition in a suitable DMD mouse disease model. Based upon our experience, we believe that the levels of NPT, iNOS, and Nox4 may be decreased in the DMD mice treated with Sirt3 inhibitor(s). Along the same lines, it is reasonable to expect that the respective levels of MTHFS, methylation, DNMT, M1 macrophages, TLR4, NLRP3, T-cells, and helper immune cells may also be decreased in the Sirt3 blocker group of the DMD mice. Such experiments might reveal that DMD disease is capable of inducing DNMTs, and HDACs (such as Sirt3), and that, in turn, can instigate the inflammatory cytokines such as TNF-α, and IL1β along with iNOS, Nox4, NPT, M1 macrophages, MMP9, and increase CD4^+^, and CD8^+^ cells. It is expected that the Sirt3 inhibitor may be able to mitigate diaphragm atrophy, and DMD disease-associated alveolar leakage. Although the use of a Sirt3 inhibitor makes sense, however; the use of a DNMT inhibitor is also justified, as described earlier [65]. Since the FOCM cycle produces homocysteine (Hcy), so measuring the Hcy levels in a genetic model of homocysteinemia, and hypermethylation may also be undertaken [51,66]. To further dissect the role of MMP9, the mice may be cross-bred with MMP9-KO strain [16,17]. These genetic approaches could further pinpoint the epigenetic mechanisms in DMD disease-associated myopathy, and lung dysfunction, and their mitigation by the inhibitors.

## 5. Role(s) of iNOS and TLR2/4

The induction of macrophage population and iNOS are the hallmarks of inflammation [67]. The iNOS utilizes BH4 to generate NO, but unfortunately during inflammation, and oxidative stress response the ROS production creates peroxynitrite, and nitrotyrosine [68]. These events cause depletion of BH4 and generate neopterin (NPT) leading to the activation of MMPs to initiate the tissue remodeling process. This change is downstream from epigenetics programming or induction. Going further, in establishing the causative role(s) of iNOS in diaphragm dysfunction and lung injury, it is important to inhibit the iNOS in the DMD disease mouse model. Previously, we showed that inhibition of iNOS helped mitigated the vascular remodeling in hyperhomocysteinemic (HHcy) mice [67]. We have also shown the role of iNOS in lung ischemia-reperfusion injury [18]. Thus, it would be good to demonstrate the role of iNOS in the induction of TNF-α, IL1β, Nox4, NPT, M1 macrophages, MMP9, and increases CD4^+^ and CD8^+^ cells in DMD disease progression. The iNOS inhibitor is expected to mitigate these levels, diaphragm atrophy, and alveolar leakage. It will also be important to determine the contribution of macrophage iNOS in the induction of TNF-α, IL1β, Nox4, NPT, M1 macrophages, and MMP9, and the levels of CD4^+^ and CD8^+^ cells during DMD pathogenesis. Because iNOS is downstream of epigenetics modifications, we anticipate no effect(s) on the levels of DNMT2, HDAC3, and MTHFS. In our previous work we have shown the induction of iNOS in lung ischemia-reperfusion injury, thus one may anticipate that iNOS will be increased in DMD disease lungs, and the skeletal muscle. Since iNOS depletes BH4, this would lead to the unavailability of endothelial eNOS, therefore, it can be assumed that there would be a decrease in endothelial function in the pulmonary artery, and hence some contribution to leakage. We hope that future experiments will determine the precise role of iNOS in pulmonary vascular remodeling during DMD disease progression. In this context, we anticipate mitigation of DMD disease induced-iNOS, and cytokine levels of TNF-α, IL1β, Nox4, NPT, M1 macrophages, MMP9, and the increase in CD4^+^, and CD8^+^ cells, diaphragm atrophy, and alveolar leakage in the DMD disease mice treated with iNOS inhibitor. There appears to be a link between dystrophin, and iNOS. Hence, to determine the role of iNOS in DMD disease, it would be wise, and important to use an iNOS blockade strategy. In addition, one can create double knockouts using DMD disease mice, and cross them with the iNOS knockout. In the double knockout mice (DMD/iNOS-/-), a clear mitigation of DMD induced-iNOS, cytokines (TNF-α, and IL1β), Nox4, NPT, M1 macrophages, MMP9, and increase in CD4^+^ and CD8^+^ cells, diaphragm atrophy, and alveolar leakage might be observed.

Again, TLR2/4 activation is part of inflammation, and oxidative stress response in various diseases including DMD disease [69,70]. Further, in DMD disease, the interaction of TLR4 with the ligand DAMP, creates an environment that stimulates cytokines production, metabolic alterations, and epigenetics programming that can lead to innate immune activation and macrophage polarization towards inflammatory phenotype [71]. Although corticosteroids help improve the mobility, and longevity of DMD patients, the mechanism is not yet fully clear [72,73,74]. In previous studies, steroid treatment in DMD patients did reduce cytotoxic/suppressor T cells including T cell infiltration in the muscle fibers resulting in reduced inflammation, and damage suggesting an important role for T cells in skeletal muscle injury [75,76]. However, whether TLR4 signaling mediates T cell priming in DMD diaphragm, and lung remains unknown. In addition, long-term steroid usage is associated with severe side effects that may hinder the quality of life in patients. It is, therefore, important to study the interaction between TLR4, and T cells in DMD disease and to investigate whether TLR4 inhibition suppresses the T cell population (CD4^+^/CD8^+^) and priming to reduce diaphragm, and lung injury. Recent studies have shown that TLR4-NLRP3-GSDMD mediates pyroptosis that causes liver injury in septic mice and contributes to tubular injury in the diabetic kidney [77,78]. Pyroptosis occurs due to the cleavage of GSDMD by caspase-1 or caspase-4, -5, and -11 via the canonical or non-canonical pathways [79]. It is therefore essential to delineate whether TLR4 activation leads to NLRP3 inflammasome formation and pyroptosis in DMD patients.

Earlier studies have shown that diaphragm fibrosis is a contributing factor to respiratory insufficiency in DMD patients [80,81]. A recent study has further revealed that changes in extracellular matrix (ECM) reorganize transverse collagen muscle fibers in the diaphragm to increase stiffness as the disease progresses [82]. It is, therefore, important to elaborate further on the role of matrix metalloproteinase-9 (MMP9) in ECM metabolism in the DMD patients’ diaphragm. We would like to re-emphasize that since TLR4 is downstream to epigenetics programming, and the iNOS pathway, it is envisaged that no change in the levels of epigenetic factors DNMT2, HDAC3, and MTHFS, and iNOS may be observed. However, the factors: TNF-α, IL1β, Nox4, NPT, M1 macrophages, MMP9, and increase in CD4^+^ and CD8^+^ cells may be affected in DMD patients and the TAK242 treated mice. In TAK242 treated mice, the reduction of inflammatory cytokines, and T cells indicate the TLR4 activation, and signaling that may be the cause of enhanced T cell proliferation in DMD lung, and diaphragm. An increase in GSDMD and caspase-1 may suggest pyroptosis in DMD patients by the canonical pathway. An increase in caspases 4, -5, and -11 might also suggest the activation of the non-canonical pathway. In short, diaphragm atrophy, and alveolar leakage in DMD mice and reduced fibrosis, and leakage following TAK242 treatment could be expected. Interestingly, if epigenetics markers DNMT, HDAC, and MTHFS are affected, it would mean that the promotor of TLR4 has certainly been epigenetically modified. If iNOS is affected, one can suggest that the promotor of iNOS is also epigenetically programmed. Since there exists heterogeneity in the TLR2/4 receptor, the blockade with TAK242 may not be specific, so to demonstrate the specific role of TLR4 in the diaphragm muscle remodeling, researchers can generate double knockouts of DMD by crossing with TLR4KO as described in our previous studies [29,30,31,32]. This will demonstrate unequivocally the role of TLR4 in the lung, and diaphragm remodeling during DMD pathogenesis.

## 6. Trans-Sulfuration and Renal Dysfunction during Hyperhomocysteinemia (HHcy)

Epigenetically governed methylation by gene writer (DNMTs) generates the homocysteine (Hcy) [83,84]. The primary function of the trans-sulfuration pathway for the kidney is to help convert Hcy into cysteine to sulfites (SO3^2−^), and sulfates (SO4^2−^) during the metabolism of methionine (Figure 3) [85]. A defect in the trans-sulfuration pathway can create homocystinuria, renal dysfunction, and cysteine stones [86]. It is known that TRPV1 regulates calcium (Ca^2+^) ion exchange in the kidney, thus a dysfunctional TRPV1 can accumulate calcium, and that may lead to calcified stones. Therefore, it would be nice to understand the mechanism(s) of renal trans-sulfuration, and generation of kidney cysteine stones. Interestingly, during epigenetically enforced hypermethylation, Hcy levels are elevated (i.e., hyperhomocysteinemia, HHcy), and HHcy is one of the causes for the formation of cysteine stones. The 3-mercaptopyruvate sulfotransferase (3-MST) generates sulfites, however; a decrease in 3-MST levels can cause cystinuria, and stones, as well. The epigenetics cycles of methylation/demethylation, acetylation/de-acetylation, and acetyl-CoA/CoA are the hallmarks of gene regulation, and mitochondrial bioenergetics, respectively (Figure 3). In brief, the defective epigenetics pathways can create homocystinuria, dysregulation of mitochondrial bioenergetics, and malfunctioning of trans-sulfuration, cysteine stone formation, and renal dysfunction [51,83,84,87,88].

## 7. Summary and Future Perspectives

We believe that dysfunctional epigenetics, DNMTs, HDACs, MTHFS, and iNOS generate NPT, and oxidative peroxynitrite/nitrosylation to activate MMP9/CD147 pathway. In this context, the M1 macrophages induce inflammasome formation via the NPT/NLRP3/TLR4 axis. Regarding the cell damage, GSDMD mediates pyroptosis, and increases immune response (↑ CD4^+^ and CD8^+^ cells), causing diaphragm dysfunction, and alveolar leakage during DMD disease pathogenesis. Therefore, it is worth to study that a treatment with epigenetic inhibitor, such as Sirt3 may mitigate diaphragm muscle atrophy, and the lung injury. We surmise that the use of Sirt3, iNOS, and TLR4 inhibitors in the lung, and diaphragm muscle remodeling during DMD, is novel as this will demonstrate the mechanistic role of progressive diaphragm, and lung associated pathology. Further, in our opinion, the mitigation of systemic remodeling of skeletal muscle that takes place in DMD disease sounds like an innovative approach. In that context, the lung specific MMP9 can be inhibited by Sirt3, iNOS, and TLR4 inhibitors. This approach can prove therapeutically novel. The oxidative peroxynitrite/nitrosylation activates MMP9 via CD147. As mentioned earlier, the M1 macrophages induce inflammasome formation via NPT/TNF-α/TLR4/NLRP3, and gasdermin D (GSDMD) mediated pyroptosis, which increases the immune response (↑ CD4^+^, and CD8^+^ cells), causing diaphragm dysfunction, and alveolar leakage during DMD, and COVID-19 infection. Since NPT levels are also robust in COVID-19 patients, the mitigation of co-morbid conditions such as long-COVID associated pulmonary dysfunction is important in understanding, and treating DMD patients, and the long-COVID patients.

## Figures and Tables

**Figure 1 ijms-24-00287-f001:**
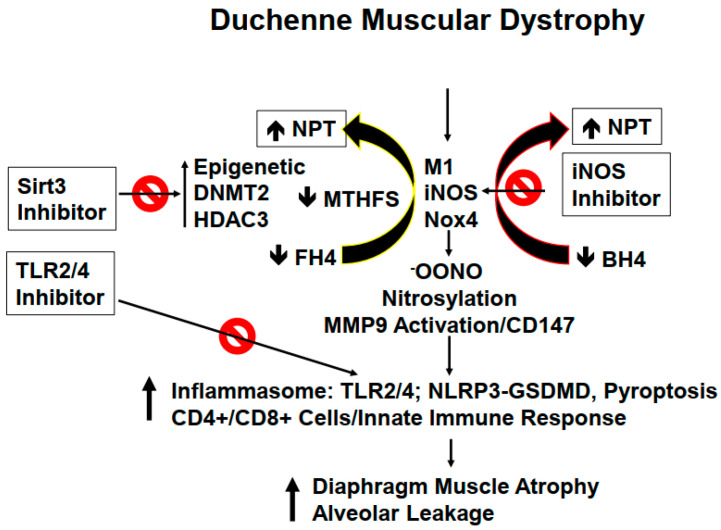
Pulmonary dysfunction caused by the mutated dystrophin protein in DMD patients. The differential interplay of various molecules in DMD patients results in severe lung injury (alveolar leakage) that is further complicated by atrophy of the diaphragm in the affected DMD patients.

**Figure 2 ijms-24-00287-f002:**
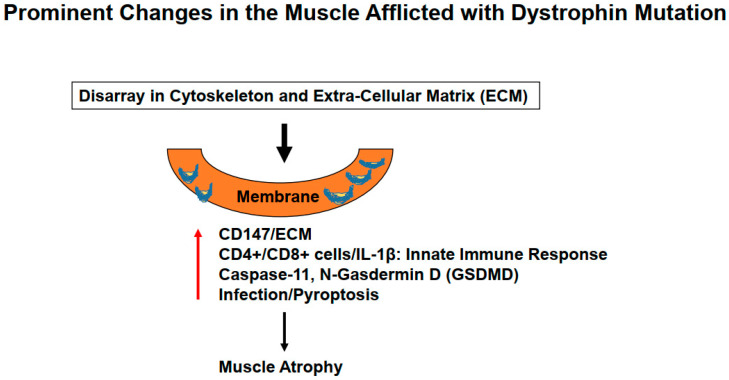
Mutation in the dystrophin protein causes derangements in the cytoskeletal leading to ECM remodeling causing activation of the innate immune system, severe inflammation, and lung injury.

**Figure 3 ijms-24-00287-f003:**
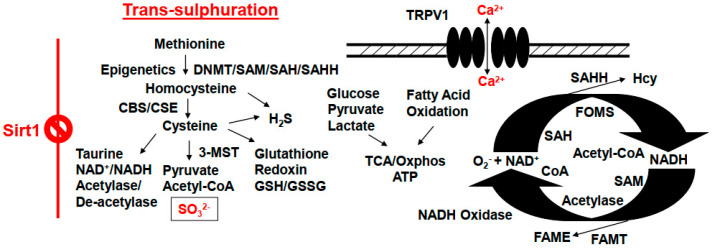
Defects in the epigenetics programming can lead to pathological outcomes by creating homocystinuria, dysregulation of mitochondrial bioenergetics, trans-sulfuration malfunctioning, formation of cysteine stones, and the onset of renal dysfunction. DNMT, DNA methyltransferase; CBS, cystathionine-β synthase; CSE, cystathionine-γ lyase; Oxphos, oxidative phosphorylation; TRPV1, transient receptor potential cation channel, also known as the capsaicin receptor and the vanilloid receptor 1; SAM, s-adenosyl-methionine; SAH, s-adenosyl homocysteine; SAHH, s-adenosyl homocysteine hydrolase; FAME, fatty acid methyl ester; FAMT, fatty acid methyltransferase; FOMS, folate 1-carbon metabolism methyl synthase; TCA, tricarboxylic acid/citric acid cycle.

## Data Availability

Not applicable.

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
