# Peer review of "COVID-19 Mimics Pulmonary Dysfunction in Muscular Dystrophy as a Post-Acute Syndrome in Patients"

_ijms, 2022, doi:10.3390/ijms24010287_

Round 1

Reviewer 1 Report

1. Plagiarism software/application ithenticate shows 18% similarity report. Can you amend the same?

2. The sentence (212 & 213) is not properly placed.

3. There are the headings 'Application of SIRT inhibitors' & 'SIRT1 can potentially.......' Why both are mentioned separately?

4. SIRT1 is beneficial in DMD - however you have indicated the application of SIRT inhibitors in DMD. Sirtuins such as 1-7 have a diverse role in DMD, therefore, do you think that is it correct to mention as SIRT inhibitors?

Author Response

Response to reviewers’ comments by the authors

Dear Editor/Reviewers:

We take this opportunity to thank you for the comments and feedback regarding our manuscript entitled “COVID-19 mimics pulmonary dysfunction in muscular dystrophy as a post-acute syndrome in patients”. As advised, below is our response to the reviewers’ comments, one-by-one. Further, per your kind advice, all the suggested changes and modifications have been incorporated in the revised manuscript. Also, relevant information, as suggested by the reviewers, has been updated in the text, and references.

Finally, we remain obliged to you/reviewers in helping us improving the quality of this manuscript further, and hope that our revised manuscript will be accepted soon without delay.

Reviewer 1

Comment 1. Plagiarism software/application ithenticate shows 18% similarity report. Can you amend the same?

Response: We thank the reviewers for the comment/suggestion, and per advice, we have tried our best to avoid most of the similarities in the text material of the revised manuscript.

 Comment 2. The sentence (212 & 213) is not properly placed.

Response: We have properly placed/corrected the 212- and 213-line sentence now. Appreciate the reviewer for pointing the error. Thank you!

 Comment 3. There are the headings 'Application of SIRT inhibitors' & 'SIRT1 can potentially.......' Why are both mentioned separately?

Response: We apologize for the oversight. In the revised version of the manuscript, we have corrected the headings. Thanks for the comment.

 Comment 4. SIRT1 is beneficial in DMD - however you have indicated the application of SIRT inhibitors in DMD. Sirtuins such as 1-7 have a diverse role in DMD, therefore, do you think that is it correct to mention as SIRT inhibitors?

Response: We thank the reviewer for the comment, and the reviewer is right that Sirtuins do have diverse roles in our body. For example, the SIRT3 encodes class III histone deacetylase that essentially represents the member of the sirtuin family, which is responsible for metabolism, however; SIRT1 is involved in inflammation as well as its role in metabolism, and so on. Thus, where body/cells’ functions are affected negatively then inhibitory approach sounds reasonable as explained in the Figure 1. Again, we thank the reviewer for the comment.

Thank you!

Reviewer 2 Report

The proposed review of Tyagi and collegues addresses an important clinical problem of recent years, whether long-term pulmonary dysfunction after COVID-19 could mimic the alterations seen in muscular dystrophy. The authors provide an extensive literature overview on the topic of Duchenne muscular dystrophy pathogenesis, with detailed description of several molecular pathways also including suggestions for future studies.

The text is overall very well written, although the paragraph between line 214 and 238 needs thorough revision and grammar corrections, as it contains sentences without meaning (line 214), duplicate words ("that that" line 218) etc.

What is really missing from the review is a concise connection of COVID-19 to DMD. Although Covid is mentioned in 6-7 sentences, only 5 references are related to Covid and none of the DMD pathomechanism descriptions include  references that directly could connect to Covid. In that context, the supposed title of the review is misleading, as Covid is not much discussed throughout the text. Also, the authors included too many merely speculative sentences (or even full paragraphs) about what could be important to investigate in future studies - in my opinion 1 such sentence for each pathogenesis would be enough.

Typos need to be revised as well ("eg. tran-sulphuration" instead of trans-suplhuration; "detective epigenetics" instead of defective, minor errors in lines 248, 252, 261, 270, etc.). Line 280: "improve ambulation" - please re-phrase.  DAMP is not listed among the abbreviations. 

Lines 200 to 208: This is a long speculative description, could more fit into the "Summary and future perspectives" chapter. The authors refer to Sirt3 inhibitors but then some drugs should be also named here.

The chapter about hyperhomocysteinaemia between lines 325 to 350 does not have any reference at all - please correct!

The list of references should have the same format for all citations, now it is very heterogenic.

The chapter title "Discussion" seems a bit odd in case of a review; it is anyway a summary of the text, I recommend to call this chapter "Summary" or "Summary and future perspectives" so that the speculative sentences could be moved to this last chapter.

Author Response

Response to reviewers’ comments by the authors

Dear Editor/Reviewers:

We again take this opportunity to thank you for the comments and feedback regarding our manuscript entitled “COVID-19 mimics pulmonary dysfunction in muscular dystrophy as a post-acute syndrome in patients”. As advised, below is our response to the reviewers’ comments, one-by-one. Further, per your kind advice, all the suggested changes and modifications have been incorporated in the revised manuscript. Also, relevant information, as suggested by the reviewers, has been updated in the text, and references.

Finally, we remain obliged to you/reviewers in helping us improving the quality of this manuscript further, and hope that our revised manuscript will be accepted soon without delay.

Reviewer 2

Comment: The proposed review of Tyagi and colleagues addresses an important clinical problem of recent years, whether long-term pulmonary dysfunction after COVID-19 could mimic the alterations seen in muscular dystrophy. The authors provide an extensive literature overview on the topic of Duchenne muscular dystrophy pathogenesis, with detailed description of several molecular pathways also including suggestions for future studies.

Response: We remain thankful to the reviewer for the kind words. Appreciated the encouragement by the reviewer.

Comment: The text is overall very well written, although the paragraph between line 214 and 238 needs thorough revision and grammar corrections, as it contains sentences without meaning (line 214), duplicate words ("that that" line 218) etc.

Response: We agree with the reviewer that paragraph between line 214 and 238 needs thorough revision and grammar corrections, as it contains sentences without meaning (line 214), duplicate words ("that that" line 218) etc. Following the reviewer’s advice/suggestion, we have thoroughly revised the paragraph, and did grammar correction also since it contained sentences without meaning. Thank you again for the comment, and we appreciate it very much.

Comment: What is really missing from the review is a concise connection of COVID-19 to DMD. Although Covid is mentioned in 6-7 sentences, only 5 references are related to Covid and none of the DMD pathomechanisms descriptions include references that directly could connect to Covid. In that context, the supposed title of the review is misleading, as Covid is not much discussed throughout the text. Also, the authors included too many merely speculative sentences (or even full paragraphs) about what could be important to investigate in future studies - in my opinion 1 such sentence for each pathogenesis would be enough.

Response: We agree with the reviewer that concise connection/relationship between COVID-19 and DMD is missing, therefore; per reviewer’s advice we have now added a few more references that deal with the connection between DMD and COVID-19. However; we also want to mention here that there are not many published studies yet that describe the role of epigenetics vis a vis putative contribution of molecules such as NPT, TNF-α, HDAC, IL-1β, CD147, and MMP9 in the pathogenesis of COVID-19 and DMD. WE hope that future studies might be able to establish the actual patho-mechanisms in the DMD and COVI-19 patients. We must state politely that ours is a review manuscript, and we do thank the reviewer for the comment though. Further, the comment is well taken, and our ongoing research work in the lab hopefully will address some of the concerns that reviewer has mentioned.  

Comment: Typos need to be revised as well ("e.g., trans-sulphuration" instead of trans-sulphuration; "detective epigenetics" instead of defective, minor errors in lines 248, 252, 261, 270, etc.). Line 280: "improve ambulation" - please re-phrase.  DAMP is not listed among the abbreviations. Lines 200 to 208: This is a long speculative description, could more fit into the "Summary and future perspectives" chapter. The authors refer to Sirt3 inhibitors but then some drugs should be also named here.

Response: We appreciate the reviewer for pinpointing errors, and thus remain thankful to the reviewer. Per reviewer’s advice, and suggestion, we now have revised the typos in body of the text as well as changed the trans-sulphuration" to trans-sulfuration, and "detective epigenetics" to defective epigenetics. Also, we have rephrased the wording for “improve ambulation" and per suggestion the abbreviation- DAMP has now been listed among the abbreviation’s list. Regarding the “speculative description”, as mentioned above, ours is a review manuscript, and we do thank the reviewer for the comment. Further, Sirt3 inhibitors or compounds names have been mentioned now in the revised version of the manuscript along with pertinent references.

Comment: The chapter about hyperhomocysteinemia between lines 325 to 350 does not have any reference at all - please correct!

Response: We apologize for this oversight, and per reviewer’s advice we have now modified “chapter about hyperhomocysteinemia” in the revised version of our manuscript. Also, we have shortened the sub-title to “Tran-sulfuration and Renal Dysfunction during Hyperhomocysteinemia (HHcy)” as suggested by another reviewer.

Thank you for pointing this oversight. The reviewer is appreciated a lot!

Comment: The list of references should have the same format for all citations, now it is very heterogenic.

Response: As per the comment of the reviewer, in revised version of our manuscript, the list of references has been corrected, and a same format for all citations has been done in the revised manuscript. We would like to thank the reviewer for suggestion/advice.

Comment: The chapter title "Discussion" seems a bit odd in case of a review; it is anyway a summary of the text; I recommend calling this chapter "Summary" or "Summary and future perspectives" so that the speculative sentences could be moved to this last chapter.

Response: The reviewer is right, and we thank the respected reviewer for the suggestion, and per reviewer’s advice, we have now changed ‘Discussion’ title to “Summary and Future Perspectives” sub-title in our revised manuscript. Thank you again.

Round 2

Reviewer 1 Report

Thanks for your changes.

The plagiarism iThenticate still shows 17%. Can you please check the manuscript and reduce plagiarism percent?

Author Response

Comment: The plagiarism iThenticate still shows 17%. Can you please check the manuscript and reduce plagiarism percent?

Reply: Many thanks to the reviewer/editor for the comment, and we do appreciate it. During the 2nd-time revision of our manuscript we have attempted to improve it further per the suggestion.

It is a 'review' manuscript, and we have included all the relevant/pertinent references (where needed) thus giving the full credit to others (the cited authors' via their references) for their work.

We have tried again to remove the same wordings/text material without changing too much of the original meaning/context (for the 2nd time revision).

Again, we want to thank the reviewer for the comment.

Reviewer 2 Report

The revised manuscript has been improved. Still, line 351 title has a typo "Tran-sulfuration" instead of "Trans-sulfuration".

Otherwise have no more concerns.

Author Response

Comment: The revised manuscript has been improved. Still, line 351 title has a typo "Tran-sulfuration" instead of "Trans-sulfuration".

Otherwise have no more concerns.

Reply: We are glad to learn that the reviewer/editor has been very encouraging about our work, and we want to thank for the same.

Regarding the typo "Tran-sulfuration" instead of "Trans-sulfuration", we sincerely apologize for the error, again. Per kind suggestion, in the 2nd-time revised manuscript, we have updated it by fixing the above typo. Thank you very much!